# Comparison of Postoperative Adjuvant Chemotherapy and Concurrent Chemoradiotherapy for FIGO2018 Stage IIIC1 Cervical Cancer: A Retrospective Study

**DOI:** 10.3390/medicina57060548

**Published:** 2021-05-29

**Authors:** Masahiro Kagabu, Takayuki Nagasawa, Shunsuke Tatsuki, Yasuko Fukagawa, Hidetoshi Tomabechi, Eriko Takatori, Yoshitaka Kaido, Tadahiro Shoji, Tsukasa Baba

**Affiliations:** Department of Obstetrics and Gynecology, Iwate Medical University School of Medicine, Shiwa, Iwate 028-3695, Japan; tnagasaw@iwate-med.ac.jp (T.N.); 412.sailing@gmail.com (S.T.); suga_yasuko@yahoo.co.jp (Y.F.); bechitomabehi@gmail.com (H.T.); takatori@iwate-med.ac.jp (E.T.); kaido0428@yahoo.co.jp (Y.K.); tshoji@iwate-med.ac.jp (T.S.); babatsu@iwate-med.ac.jp (T.B.)

**Keywords:** cervical cancer, International Federation of Gynecology and Obstetrics, postoperative adjuvant therapy, systemic chemotherapy, concurrent chemoradiotherapy

## Abstract

*Background and Objectives*: In October 2018, the International Federation of Gynecology and Obstetrics (FIGO) revised its classification of advanced stages of cervical cancer. The main points of the classification are as follows: stage IIIC is newly established; pelvic lymph node metastasis is stage IIIC1; and para-aortic lymph node metastasis is stage IIIC2. Currently, in Japan, radical hysterectomy is performed in advanced stages IA2 to IIB of FIGO2014, and concurrent chemoradiotherapy (CCRT) is recommended for patients with positive lymph nodes. However, the efficacy of CCRT is not always satisfactory. The aim of this study was to compare postoperative adjuvant chemotherapy (CT) and postoperative CCRT in stage IIIC1 patients. *Materials and Methods*: Of the 40 patients who had undergone a radical hysterectomy at Iwate Medical University between January 2011 and December 2016 and were pathologically diagnosed as having positive pelvic lymph nodes, 21 patients in the adjuvant CT group and 19 patients in the postoperative CCRT group were compared. *Results*: The 5 year survival rates were 77.9% in the CT group and 74.7% in the CCRT group, with no significant difference. There was no significant difference in overall survival or progression-free survival between the two groups. There was no significant difference between CT and CCRT in postoperative adjuvant therapy in the new classification IIIC1 stage. *Conclusions*: The results of the prospective Japanese Gynecologic Oncology Group (JGOG) 1082 study are pending, but the present results suggest that CT may be a treatment option in rural areas where radiotherapy facilities are limited.

## 1. Introduction

Cervical cancer is the seventh most common cancer in humans and the fourth most common cancer in women. According to the Agency for Research on Cancer (IARC) database in 2018, the global incidence of cervical cancer was approximately 569,000, and the global mortality was 311,000 [1]. The International Federation of Gynecology and Obstetrics (FIGO) revised the staging system for carcinoma of the uterine cervix in 2018 [2]. The major change point from the prior 2014 FIGO staging system is the incorporation of lymph node status into stage III disease staging. Stage IIIC was established for cases with lymph node metastasis, with stage IIIC1 for pelvic lymph node metastasis only and stage IIIC2 for para-aortic lymph node metastasis. Patients with early-stage cervical cancer require radical hysterectomy with pelvic lymphadenectomy in Japan. When risk factors for recurrence (positive surgical margins, positive lymph nodes, etc.) are identified after surgery, concurrent chemoradiotherapy (CCRT) has been performed. However, postoperative radiation therapy can cause severe adverse events for a long time. It is particularly distressing for adverse events in the gastrointestinal tract, significantly reducing the quality of life of patients [3,4]. Matoda et al. evaluated the efficacy of adjuvant CT with irinotecan (CPT-11)/nedaplatin (NDP) in patients with positive postoperative lymph nodes [5]. Their report showed that 2 year and 5 year relapse-free survival (RFS) rates were 87.1% and 77.2%. Furthermore, 22.5% of patients relapsed during the follow-up period; 12.9% of whom died of the disease. The 5 year overall survival (OS) rate in this study was 86.5%. Takekuma et al. reported a phase II study of adjuvant CT with paclitaxel (PTX) and NDP for uterine cervical cancer with lymph node metastasis [6]. Their study showed that the 2 year RFS and OS were 79% and 93.5%, respectively. All of these were reports of a small number of cases, and thus, their generalizability remains unclear. Based on the results, a phase III trial of CCRT and CT is currently underway (JGOG1082). In the present single-center study, the postoperative adjuvant CT was retrospectively compared with postoperative CCRT in FIGO stage IIIC1 to provide a validating case-series sample.

## 2. Materials and Methods

### 2.1. Patients

A total of 40 patients with cervical cancer were treated from 2012 to 2016 at Iwate Medical University Hospital. All patients underwent radical hysterectomy (type III) and pelvic lymphadenectomy before CCRT or CT. The patients were FIGO stage IIIC1 (positive pelvic nodes).

### 2.2. Treatment

#### 2.2.1. CCRT

Postoperative radiotherapy consisted of external beam radiotherapy (EBRT). EBRT was delivered with anteroposterior- and posteroanterior-opposed beams generated by an X-ray accelerator with an energy of 10 MV at a distance of 100 cm. The superior margin of the external radiation field was located at the top of the L4-5 vertebrae, and the inferior margin was located at the inferior border of the obturator foramen. The lateral margin was 1.5–2 cm lateral to the widest margin of the bony pelvis. Whole pelvic irradiation was delivered at 2 Gy/fraction for 5 fractions/week, for a total of 25 fractions (50 Gy). CT was given intravenously weekly during the course of radiotherapy. The first administration was performed on the first day of radiotherapy. The dose of cisplatin (CDDP) was 40 mg/m^2^. The median number of cycles per patient with CDDP was 5 (range: 2–5). CDDP was given for >180 min. Renal function and blood counts were assessed before each cycle. Adjuvant treatment was started approximately 7 weeks later (median: 50 days, range: 35–71).

#### 2.2.2. Chemotherapy

Chemotherapy was administered intravenously. Five patients received CPT-11/NDP (CPT-11 60 mg/m^2^ on days 1 and 8, and NDP 80 mg/m^2^ on day 1), and the cycles were repeated every 28 days. CPT-11 was given over 1.5 h, and NDP was given over 1 h. The median number of cycles per patient with CPT-11/NDP was 6 (range: 5–6). Four patients received CPT-11/CDDP (CPT-11 60 mg/m^2^ on days 1 and 8 and 15, and CDDP 60 mg/m^2^ on day 1), and the cycles were repeated every 28 days. CPT-11 was given over 1.5 h, and cisplatin was given over 1.5 h. The median number of cycles per patient with CPT-11/CDDP was 5 (range: 4–6). Four patients received PTX/NDP (PTX 175 mg/m^2^ and NDP 80 mg/m^2^ on day 1), and the cycles were repeated every 28 days. Paclitaxel was given over 3 h, and nedaplatin was given over 1 h. The median number of cycles per patient with PTX/NDP was 5 (range: 3–5). Five patients received docetaxel (DTX)/ carboplatin (CBDCA) (DTX 70 mg/m^2^ and CBDCA AUC5 on day 1), and the cycles were repeated every 21 days. DTX was given over 1 h, and CBDCA was given over 1 h. The median number of cycles per patient with DTX/CBDCA was 6 (range: 3–6). Four patients received PTX/CBDCA (PTX 175 mg/m^2^ and CBDCA AUC6 on day 1), and the cycles were repeated every 21 days. PTX was given over 3 h, and carboplatin was given over 1 h. The median number of cycles per patient with PTX/CBDCA was 6 (range: 5–6). Adjuvant treatment was started approximately 6 weeks later (median: 42.5 days, range: 25–87).

### 2.3. Statistical Analysis

The survival analysis was based on the Kaplan–Meier method, and the results were compared using the log-rank test. The recurrence rates were compared using Fisher’s exact test. Significance was set at *p* < 0.05. The Prism 8.0 software program (GraphPad Software Inc., San Diego, CA, USA) was used for all statistical analyses.

## 3. Results

### 3.1. Patients’ Characteristics

Patient characteristics are shown in Table 1. Of these, 19 patients were treated with CCRT, and the other 21 patients were treated with CT. In the CCRT group, the mean age of the patients was 42 years (range 28–67 years). Eleven patients were T1b (UICC-TNM), 3 patients were T2a, and 5 patients were T2b. Eighteen patients had squamous cell carcinoma (SCC), and 1 patient had adenocarcinoma. In the CT group, the mean age of the patients was 44 years (range 32–61 years). Fourteen patients had T1b, and 7 patients had T2b. Twelve patients had SCC, 8 patients had adenocarcinoma, and 1 patient had small-cell carcinoma. All patients in this study had a negative surgical margin. There were no significant differences in terms of age, number of metastatic lymph nodes, and time to start adjuvant treatment between the groups. On the other hand, there were a significant difference in the clinical stage and histological distribution.

### 3.2. Treatment Outcomes

As shown in Figure 1, when the CCRT group was compared with the CT group, relapse-free survival rate (RFS) (log-rank: *p* = 0.9751, Hazard ratio [HR]: 1.020, 95% confidence interval [CI]: 0.2885–3.605), and overall survival rate (OS) (log-rank: *p* = 0.6868, HR: 0.7266, 95% CI: 0.1233–4.281) were not significantly different. As shown in Figure 2, when the CCRT with single-lymph-node metastasis group was compared with the CT with single-lymph-node metastasis group, PFS (log-rank: *p* = 0.6539, HR: 0.6481, 95% CI: 0.087–4.819) and OS (log-rank: *p* = 0.9795, HR: 0.9706, 95% CI: 0.086–10.84) were not significantly different. As shown in Figure 3, when the CCRT with multiple-lymph-node metastasis group was compared with the CT with multiple-lymph-node metastasis group, PFS (log-rank: *p* = 0.7173, HR: 0.7617, 95% CI: 0.1633–3.553) and OS (log-rank: *p* = 0.8911, HR: 0.8607, 95% CI: 0.081–9.066) were not significantly different. In addition, no statistically significant difference RFS and OS were found between CCRT and CT in parametrium invasion (PFS log-rank: *p* = 0.8087, HR: 0.7872, 95% CI: 0.1093–5.668, OS log-rank: *p* = 0.6084, HR: 0.4933, 95% CI: 0.0259–9.394, Figure 4).

As shown in Table 2, the recurrence rate of the CCRT group was 26.3%, and the recurrence rate of the CT group was 33.3%. The recurrence rate was not significantly different between the CCRT and CT groups (*p* = 0.7365). In the CCRT Group, 4 patients recurred outside the pelvis, and 1 patient recurred both in the pelvis and outside the pelvis. In the CT group, 1 patient recurred in the pelvis, 3 patients recurred outside the pelvis, and 3 patients recurred both in the pelvis and outside the pelvis.

### 3.3. Adverse Events

As shown in Table 3, there were no treatment-related deaths recorded. According to the National Cancer Institute Common Terminology Criteria for Adverse Events, version 4.0, 12 patients in the CT group had neutropenia of grade 3 or 4. In the CCRT group, there were two acute toxicity (vomiting and diarrhea) of grade 3 or 4. On the other hand, there were two sever late toxicity (small intestinal obstruction and skin infection) in the CCRT group.

## 4. Discussion

This paper is the first comparison of CCRT and CT in FIGO2018 stage IIIC1 cervical cancer. The present study found no significant differences in OS and PFS between CCRT and CT for cervical cancer patients with FIGO stage IIIC1. Adjunctive CT after radical hysterectomy was shown to be an effective treatment.

In the current retrospective study of CT, the 4 year PFS and 4 year OS were 71.3% and 77.9%, respectively. Radiotherapy or CCRT have been performed in postoperative cervical cancer patients with lymph node metastasis. CCRT has already been shown to significantly meliorate prognosis in comparison to radiotherapy alone. The GOG109/AWOG87-97 trial was a prospective study that explored the efficacy of CCRT for postoperative cervical cancer patients with risk factors for recurrence. GOG109 showed that the estimated 4 year PFS and OS were 80% and 81%, respectively [7]. Approximately 90% of patients had positive lymph nodes in GOG109. On the other hand, 90% of the patients had SCC without small-cell carcinoma. In the current study, one patient had small-cell carcinoma. Small-cell carcinoma has a poorer prognosis than other histological types of cervical cancer [8]. Liao et al. have shown that the prognostic factors were FIGO2014 stage, tumor mass size, lymph node metastasis, and depth of stromal invasion [9]. Wang et al. showed that stage and lymph node metastasis were prognostic factors [10]. CT was as effective as CCRT, even though it included histological types with poor prognoses. Several retrospective studies of postoperative adjuvant therapy alone have shown that only systemic CT as postoperative adjuvant therapy can achieve similar or better results compared to RT/CCRT [11,12,13,14]. Matsuo et al. reported that systemic CT can be expected to affect a postoperative treatment such as CCRT in node-positive FIGO2008 stage IB-IIB cervical cancer [15]. Considering the high number of high-risk factors (tumor size > 4 cm, adenocarcinoma) in the CT group, the benefit of CT can be seen in the present study.

In the present study, because the CT regimen was reviewed on a case-by-case basis, the CT regimen was not standardized. Various regimens for postoperative adjuvant CT have been investigated in cervical cancer. Takekuma et al. reported a phase II study of adjuvant CT with PTX and NDP for uterine cervical cancer with LN metastasis [6]. This study showed that the estimated 2 year RFS and OS were 79% and 93.5%, respectively. Matoda et al. reported a phase II study of adjuvant CT with CPT-11 and NDP for uterine cervical cancer with LN metastasis [5]. Their study showed that the estimated 2 year and 5 year RFS were 87% and 77.2%, respectively, and the 5 year OS rate was 86.5%. The present study showed a similar result, with estimated 2 year RFS and OS of 78.5% and 93.7%, respectively. On the other hand, Ishikawa et al. reported a retrospective study of neuroendocrine tumors of the uterine cervix examining the prognosis according to the new 2018 staging system, comparing outcomes for different chemotherapeutic regimens and histopathological subtypes [12]. They showed that response rates to etoposide-platinum or irinotecan-platinum regimens were 43.8% but only 12.9% to a taxane-platinum regimen. Nevertheless, Matsuo et al. reported that CT alone is likely insufficient for local control, and adding pelvic irradiation to systemic CT is recommended in this subgroup [15]. In the present study, there were more pelvic recurrences in the CT group than in the CCRT group. Huang et al. reported a phase III study of sequential chemoradiation (SCRT) versus CCRT or RT alone in adjuvant treatment after hysterectomy for cervical cancer [16]. They showed that SCRT provided higher DFS and lower risk than RT, but there was a significant difference from CCRT. Based on the results of this study alone, we cannot conclude that SCRT is no different from CCRT in terms of therapeutic efficacy. Adverse events of postoperative CCRT are important for patient quality of life. It is expected that there be serious gastrointestinal toxicity due to postoperative CCRT [3,4]. In this study, there is one case of serious gastrointestinal toxicity (Table 3). Although infrequent, it is necessary to be alert to the occurrence of late adverse events for postoperative CCRT on a permanent basis. The limitations of this study are that it was a retrospective study with a small sample size. Currently, the phase III trial of adjuvant CT versus CCRT for postoperative cervical cancer is underway (JGOG1082). The strength of the present study is that the quality of surgery as a confounding factor can be avoided by a single-center study. Furthermore, the CT regimen was not standardized in the present study. However, a case-by-case review of the regimen may have provided good results. Although future studies are needed, tailored CT as in ovarian cancer rather than centralized CCRT may prolong the prognosis of cervical cancer patients.

## 5. Conclusions

The present results showed that there was no significant difference in efficacy between CT and CCRT for postoperative treatment in FIGO2018 stage IIIC1 cervical cancer. Moreover, these results suggested that there was no significant difference in efficacy between CT and CCRT for postoperative treatment of parametrium-positive patients in FIGO2018 stage IIIC1 cervical cancer. Systemic CT has potential as a postoperative treatment choice for FIGO2018 stage IIIC1 cervical cancer.

## Figures and Tables

**Figure 1 medicina-57-00548-f001:**
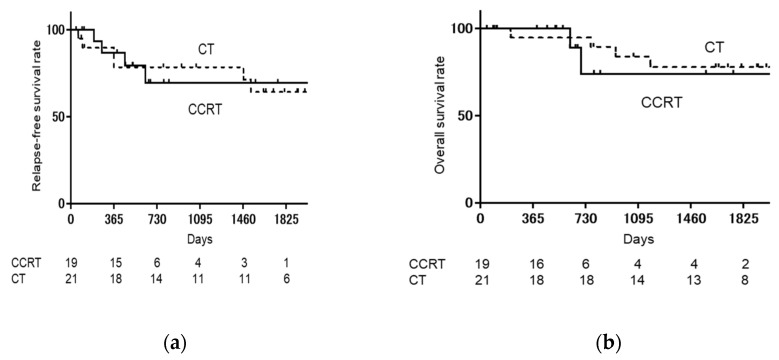
Kaplan–Meier analysis of relapse-free survival (RFS) and overall survival (OS). (**a**) Kaplan–Meier analysis of relapse-free survival (RFS); (**b**) Kaplan–Meier analysis of overall survival (OS). The CCRT group is compared with the CT group, and RFS (log-rank: *p* = 0.9751, HR: 1.020, 95% CI: 0.2885–3.605) and OS (log-rank: *p* = 0.6868, HR: 0.7266, 95% CI: 0.1233–4.281) are not significantly different. CCRT: concurrent chemoradiotherapy, CT: chemotherapy.

**Figure 2 medicina-57-00548-f002:**
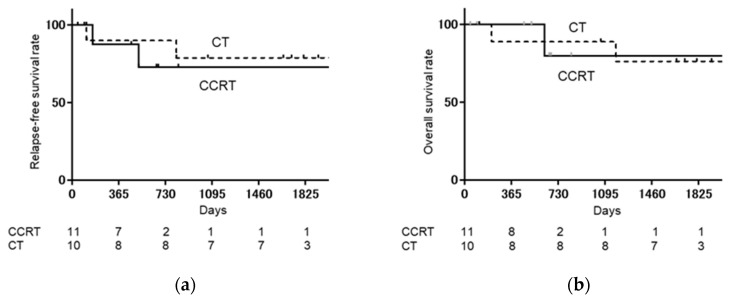
Kaplan–Meier analysis of relapse-free survival (RFS) and overall survival (OS) with single-lymph-node metastasis. (**a**) Kaplan–Meier analysis of relapse-free survival (RFS) with single-lymph-node metastasis. (**b**) Kaplan–Meier analysis of overall survival (OS) with single-lymph-node metastasis. The CCRT with single-lymph-node metastasis group is compared with the chemotherapy with single-lymph-node metastasis group, and there are no significant differences in RFS (log-rank: *p* = 0.6539, HR: 0.6481, 95% CI: 0.087–4.819) and OS (log-rank: *p* = 0.9795, HR: 0.9706, 95% CI: 0.086–10.84).

**Figure 3 medicina-57-00548-f003:**
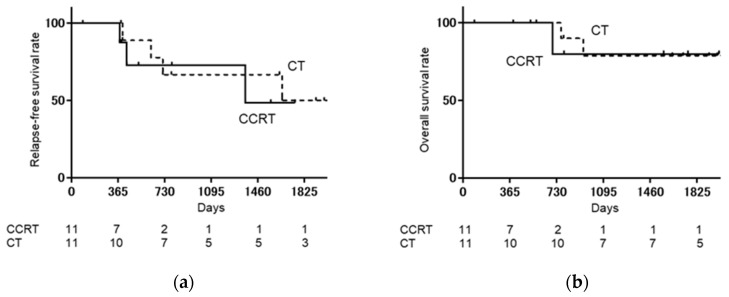
Kaplan–Meier analysis of relapse-free survival (RFS) and overall survival (OS) with multiple-lymph-node metastases. (**a**) Kaplan–Meier analysis of relapse-free survival (RFS) with multiple-lymph-node metastases; (**b**) Kaplan–Meier analysis of overall survival (OS) with multiple-lymph-node metastases. The CCRT with multiple-lymph-node metastasis group is compared with the chemotherapy with multiple-lymph-node metastasis group; there are no significant differences in RFS (log-rank: *p* = 0.7173, HR: 0.7617, 95% CI: 0.1633–3.553) and OS (log-rank: *p* = 0.8911, HR: 0.8607, 95% CI: 0.081–9.066) between them.

**Figure 4 medicina-57-00548-f004:**
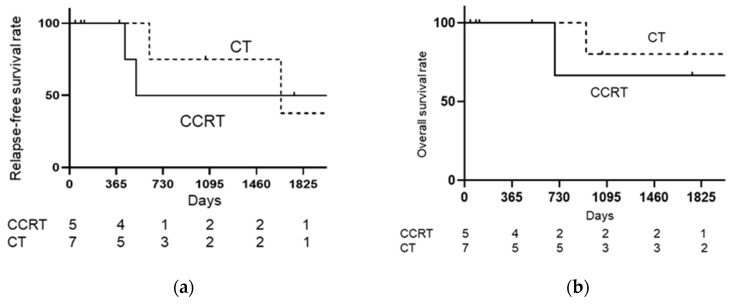
Kaplan–Meier analysis of relapse-free survival (RFS) and overall survival (OS) with pathological parametrium invasion. (**a**) Kaplan–Meier analysis of relapse-free survival (RFS) with pathological parametrium invasion; (**b**) Kaplan–Meier analysis of overall survival (OS) with pathological parametrium invasion. The CCRT with pathological parametrium invasion group is compared with the chemotherapy with pathological parametrium invasion group; there are no significant differences in RFS (log-rank: *p* = 0.8087, HR: 0.7872, 95% CI: 0.1093–5.668) and OS (log-rank: *p* = 0.6084, HR: 0.4933, 95% CI: 0.0259–9.394) between them.

**Table 1 medicina-57-00548-t001:** Patient characteristics.

	CCRT (*n* = 19)	CT (*n* = 21)	
Age, years			
Median (range)	42 (28–67)	44 (32–61)	
TNM stage			
T1B	11	14	*p* = 0.04
tumor size < 2 cm	2	0	
2–3.9 cm	5	6	
>4 cm	4	8	
T2A	3	0	
T2B	5	7	
Histology			
squamous cell carcinoma	18	12	
adenocarcinoma	1	8	*p* = 0.006
small-cell carcinoma	0	1	
Number of metastatic lymph nodes			
1	11	10	
2 or more	8	11	
Regimen of CT			
CPT11/NDP		5	
CPT11/CDDP		4	
PTX/NDP		4	
DTX/CBDCA		5	
PTX/CBDCA		3	
Time to start adjuvant treatment			
Median (range)	50 (35–71)	42 (25–87)	*p* = 0.4514

CCRT: concurrent chemoradiotherapy, CT: chemotherapy.

**Table 2 medicina-57-00548-t002:** Site of recurrence.

	CCRT	CT
Locoregional		
Intrapelvic lymph nodes		1
Distant		
Para-aortic lymph nodes	3	3
Lung	1	
Both		
Vaginal, Para-aortic lymph nodes	1	
Intrapelvic and Para-aortic lymph nodes		2
Lung, Intrapelvic lymph nodes		1

**Table 3 medicina-57-00548-t003:** Adverse events (Grade3/4).

	CCRT	CT
Acute toxicities		
Hematological toxicities		
Neutropenia		12
Anemia		0
Thrombocytopenia		0
Nonhematological toxicities		
Nausea	1	
Vomiting		
Diarrhea	1	
Late toxicities		
Small intestinal obstruction	1	
Skin infection	1	

## Data Availability

Not applicable.

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
