# Peer review of "Comparison of Postoperative Adjuvant Chemotherapy and Concurrent Chemoradiotherapy for FIGO2018 Stage IIIC1 Cervical Cancer: A Retrospective Study"

_medicina, 2021, doi:10.3390/medicina57060548_

Round 1
Reviewer 1 Report
More references needed to scientifically support the study
generally
Author Response
Reviewer 1
“More references needed to scientifically support the study.”
Thank you for your comment.
I have added new data and some references to this manuscript.

Reviewer 2 Report
This paper about 'Comparison of postoperative adjuvant chemotherapy and concurrent chemoradiotherapy for FIGO2018 stage IIIC1 cervical cancer: A retrospective study.' is interesting theme.
 CCRT is used as adjuvant therapy for cervical cancer. But the adverse events are sometimes irreversible and disastrous. It is very important to discuss whether changing CCRT to chemotherapy has the same effect. In that point, the theme of this manuscript is important.
But I have some questions for this manuscript. At present, your manuscript does not have a strong impact. I will write my suggestion for improvement, so please consider it.
Major comments
No.1
Although there are many patients with IB2, IB3 (FIGO2018) and adenocarcinoma in the CT group, the good treatment results in the CT group are findings that support the usefulness of CT. On the other hand, since this study is about adjuvant therapy after surgery, I think it is better to use the pathological stage for analysis rather than the clinical stage. I would like to know the analysis using the pathological stage.
Also, all patients have a negative margin, but is it also negative for parametrium? If there are positive cases of parametrium, I would like to know the analysis results focusing on that point.
No.2
As you mentioned, the adverse events of radiation therapy also include late effects. It can also be difficult to treat at times. The comparison between radiation therapy and chemotherapy should also compare the extent of each adverse event. It is also secondary endpoint in JGOG1082.
I would like to see CCRT and CT-related adverse events in the 40 patients included in your study. I have experienced severe adverse events at CCRT after radical hysterectomy. Your opinion that CT is an alternative to CCRT may be strongly supported by discussing adverse events. And you should add a reference on the comparison of CT and CCRT adverse events to the ‘Introduction’.
No.3
You wrote ‘The first administration was performed on the first day of radiotherapy. The median 87 dose of NDP was 30 mg/m2 (range: 10-45 mg/m2). The number of cycles per patient with 88 NDP was 5. NDP was given for >180 min. The dose of CDDP was 40 mg/m2. The number 89 of cycles per patient with CDDP was 5. CDDP was given for >180 min.’ in ‘2.2.2. chemotherapy line 87-90. Is this a description of CCRT? In any case, I would like information such as the number of courses not only for CDDP and NDP but also for other chemotherapy regimens.
And I want to know how many days after surgery CCRT and CT were started. If there is a difference in the start date between the two groups, that point should be taken into consideration when analyzing.
Minor comments
No.1
You use the expression ‘5% of patients had small cell carcinoma’ in line166. However, this is an exaggerated expression. I think the expression "one small cell carcinoma is included" is appropriate.
No.2
I think it is better to describe the number of patients in the figure1-3.
Author Response
Major comments
No.1
Although there are many patients with IB2, IB3 (FIGO2018) and adenocarcinoma in the CT group, the good treatment results in the CT group are findings that support the usefulness of CT. On the other hand, since this study is about adjuvant therapy after surgery, I think it is better to use the pathological stage for analysis rather than the clinical stage. I would like to know the analysis using the pathological stage.
Also, all patients have a negative margin, but is it also negative for parametrium? If there are positive cases of parametrium, I would like to know the analysis results focusing on that point.
Thank you for your comment.
I have added results of analysis for positive cases of parametrium and Fig 4 to this manuscript.
No.2
As you mentioned, the adverse events of radiation therapy also include late effects. It can also be difficult to treat at times. The comparison between radiation therapy and chemotherapy should also compare the extent of each adverse event. It is also secondary endpoint in JGOG1082.
I would like to see CCRT and CT-related adverse events in the 40 patients included in your study. I have experienced severe adverse events at CCRT after radical hysterectomy. Your opinion that CT is an alternative to CCRT may be strongly supported by discussing adverse events. And you should add a reference on the comparison of CT and CCRT adverse events to the ‘Introduction’.
Thank you for your comment.
I have added data of adverse evets and Table 2 to this manuscript.
And I have added a reference on the comparison of CT and CCRT adverse events to the ‘Introduction’ and “Discussion”.
No.3
You wrote ‘The first administration was performed on the first day of radiotherapy. The median 87 dose of NDP was 30 mg/m2 (range: 10-45 mg/m2). The number of cycles per patient with 88 NDP was 5. NDP was given for >180 min. The dose of CDDP was 40 mg/m2. The number 89 of cycles per patient with CDDP was 5. CDDP was given for >180 min.’ in ‘2.2.2. chemotherapy line 87-90. Is this a description of CCRT? In any case, I would like information such as the number of courses not only for CDDP and NDP but also for other chemotherapy regimens.
Thank you for your comments.
I corrected it.
And I have added information of chemotherapy regimens.
And I want to know how many days after surgery CCRT and CT were started. If there is a difference in the start date between the two groups, that point should be taken into consideration when analyzing.
Thank you for your comment.
I have added data in Table 1 to this manuscript.
Minor comments
No.1
You use the expression ‘5% of patients had small cell carcinoma’ in line166. However, this is an exaggerated expression. I think the expression "one small cell carcinoma is included" is appropriate.
Thank you for your comments.
I corrected it.
No.2
I think it is better to describe the number of patients in the figure1-3.
Thank you for your comment.
I have added number of patients in the Fig1-4.
Round 2
Reviewer 2 Report
You have corrected the manuscript better for the comments.
I have one additional comment.
Minor comment
Regarding the treatment of parametrium-positive patients, you should include the interpretation in the conclusion.